Emergence of colistin resistance and characterization of antimicrobial resistance and virulence factors of Aeromonas hydrophila, Salmonella spp., and Vibrio cholerae isolated from hybrid red tilapia cage culture

Thaotumpitak Varangkana 1
Sripradite Jarukorn 2
Atwill Edward R. 3
Jeamsripong Saharuetai saharuetai.j@gmail.com 1
1 Research Unit in Microbial Food Safety and Antimicrobial Resistance, Department of Veterinary Public Health, Faculty of Veterinary Science, Chulalongkorn University , Bangkok , Thailand
2 Department of Social and Applied Science, College of Industrial Technology, King Mongkut’s University of Technology North Bangkok , Bangkok , Thailand
3 Department of Population Health and Reproduction, School of Veterinary Medicine, University of California , Davis , United States of America
Saki Morteza
Electronic publication date: 2023 Feb 23
Publication date: 2023
Volume: 11
Electronic Location ID: e14896
Received 2022 Nov 16; Accepted 2023 Jan 24
Copyright: ©2023 Thaotumpitak et al.
Copyright year: 2023
Copyright holder: Thaotumpitak et al.
License: This is an open access article distributed under the terms of the Creative Commons Attribution License, which permits unrestricted use, distribution, reproduction and adaptation in any medium and for any purpose provided that it is properly attributed. For attribution, the original author(s), title, publication source (PeerJ) and either DOI or URL of the article must be cited.
License URL: https://creativecommons.org/licenses/by/4.0/

Keywords: Aeromonas hydrophila, Antimicrobial resistance, Colistin, Salmonella, Vibrio cholerae, Hybrid red tilapia

Funding: The University of California, Davis, USA A19-4577-S001 Chulalongkorn University fund (Ratchadaphiseksomphot Endowment Fund) National Research Council of Thailand NRCT5-RGJ63001-017 This study was funded by the University of California, Davis, USA (A19-4577-S001), and the 90th anniversary of Chulalongkorn University fund (Ratchadaphiseksomphot Endowment Fund). Varangkana Thaotumpitak is a recipient of the Royal Golden Jubilee Ph.D. program, which was supported by the National Research Council of Thailand (NRCT5-RGJ63001-017). The funders had no role in study design, data collection and analysis, decision to publish, or preparation of the manuscript.

==============================
Background

Tilapia is a primary aquaculture fish in Thailand, but little is known about the occurrence of antimicrobial resistance (AMR) in Aeromonas hydrophila, Salmonella spp., and Vibrio cholerae colonizing healthy tilapia intended for human consumption and the co-occurrence of these AMR bacteria in the cultivation water.

Methods

This study determined the phenotype and genotype of AMR, extended-spectrum β-lactamase (ESBL) production, and virulence factors of A. hydrophila, Salmonella spp., and V. cholerae isolated from hybrid red tilapia and cultivation water in Thailand. Standard culture methods such as USFDA’s BAM or ISO procedures were used for the original isolation, with all isolates confirmed by biochemical tests, serotyping, and species-specific gene detection based on PCR.

Results

A total of 278 isolates consisting of 15 A. hydrophila, 188 Salmonella spp., and 75 V. cholerae isolates were retrieved from a previous study. All isolates of A. hydrophila and Salmonella isolates were resistance to at least one antimicrobial, with 26.7% and 72.3% of the isolates being multidrug resistant (MDR), respectively. All A. hydrophila isolates were resistant to ampicillin (100%), followed by oxytetracycline (26.7%), tetracycline (26.7%), trimethoprim (26.7%), and oxolinic acid (20.0%). The predominant resistance genes in A. hydrophila were mcr-3 (20.0%), followed by 13.3% of isolates having floR, qnrS, sul1, sul2, and dfrA1. Salmonella isolates also exhibited a high prevalence of resistance to ampicillin (79.3%), oxolinic acid (75.5%), oxytetracycline (71.8%), chloramphenicol (62.8%), and florfenicol (55.3%). The most common resistance genes in these Salmonella isolates were qnrS (65.4%), tetA (64.9%), blaTEM (63.8%), and floR (55.9%). All V. cholerae isolates were susceptible to all antimicrobials tested, while the most common resistance gene was sul1 (12.0%). One isolate of A. hydrophila was positive for int1, while all isolates of Salmonella and V. cholerae isolates were negative for integrons and intSXT. None of the bacterial isolates in this study were producing ESBL. The occurrence of mcr-3 (20.0%) in these isolates from tilapia aquaculture may signify a serious occupational and consumer health risk given that colistin is a last resort antimicrobial for treatment of Gram-negative bacteria infections.

Conclusions

Findings from this study on AMR bacteria in hybrid red tilapia suggest that aquaculture as practiced in Thailand can select for ubiquitous AMR pathogens, mobile genetic elements, and an emerging reservoir of mcr and colistin-resistant bacteria. Resistant and pathogenic bacteria, such as resistance to ampicillin and tetracycline, or MDR Salmonella circulating in aquaculture, together highlight the public health concerns and foodborne risks of zoonotic pathogens in humans from cultured freshwater fish.

Introduction

Thailand is one of the main producers of global freshwater fish production. Tilapia is highlighted as the main cultivated fish in Thailand with a production of at least 200,000 tons per year (FAO, 2020). The increased demand for tilapia leads to the intensification of fish farming, and this circumstance can make fish more susceptible to bacterial and viral infection. Aeromonas spp. and Vibrio cholerae are native inhabitants of the aquatic environment. A. hydrophila is the main causative agent of motile Aeromonas septicemia (MAS), while V. cholerae is the normal flora of freshwater fish and is postulated as an opportunistic fish pathogen (Halpern & Izhaki, 2017; Nicholson et al., 2020). These two bacteria play a particular role in foodborne pathogens. A. hydrophila has been implicated in foodborne diarrheal outbreaks and the presence of V. cholerae in freshwater fish has been associated with cholera in human cases (Liu, Whitehouse & Li, 2018; Elimian et al., 2019). Salmonella is one of the top five bacterial pathogens that causes foodborne and waterborne diseases in humans (Lee & Yoon, 2021). Although Salmonella is harmless in fish, it has been associated with many foodborne diseases related to freshwater fish (Liu, Whitehouse & Li, 2018). For example, human infection with non-typhoidal Salmonella (NTS) associated with contaminated fish can cause gastroenteritis, potential bacteremia and potential life-threatening infections (Akinyemi, Ajoseh & Fakorede, 2021).

Antimicrobial resistance (AMR) is of increasing concern due to its impact on therapeutic options, resulting in failure of treatment and public health impacts. AMR bacteria found in fish and aquatic environments can originate from human activities, land-based agriculture, and aquaculture (Bollache et al., 2019). Resistant bacteria from the environment can contribute to AMR in aquaculture and possibly in humans as an important One Health issue. However, AMR occurrence in fish and the surrounding aquatic environments is poorly studied. Extended-spectrum β-lactamase (ESBL)-producing bacteria are resistant to most penicillins and cephalosporins, and can also be co-resistant to other antimicrobial groups such as fluoroquinolones and colistin (Zhang et al., 2019; Gawish et al., 2021). Multidrug resistant (MDR) A. hydrophila, Salmonella spp. and V. cholerae isolated from fish have been documented (Saharan, Verma & Singh, 2020), which can elevate the risk of co-infection with MDR and ESBL-producing bacteria. Colistin is a last-line antimicrobial reserved for the treatment of serious gram-negative bacterial infections in humans. In livestock, colistin is used primarily for the treatment of diarrhea and is currently banned in many countries, including Thailand due to the emergence of plasmid-borne colistin resistance genes (mcr) reported in meat and poultry products that indicate potential transferable resistance (Pungpian et al., 2021). The mcr genes are commonly detected in Enterobacterales, including E. coli and Salmonella, and are occasionally found in other zoonotic gram-negative bacteria such as A. hydrophila, and Klebsiella pneumonia (Elbediwi et al., 2019; Luo, Wang & Xiao, 2020). The global dissemination of the mcr genes among these pathogens is highly concerning, because they can be potentially transmitted from animals to humans, resulting in unsuccessful gram-negative bacterial infection treatment. Furthermore, the presence of resistance to colistin in fish has been reported (Liu et al., 2020; Saharan, Verma & Singh, 2020), with a recent study indicating that the consumption of fish products was a risk factor for infection by bacteria harboring the mcr-1 gene among healthy Chinese (Lv et al., 2022).

Quinolones are effective antimicrobials for the treatment of gram-negative bacterial infection. Their analogs have been applied to both humans and aquatic animals. For example, ciprofloxacin is commonly used for the treatment of gastrointestinal and urinary tract infections in humans, while oxolinic acid and enrofloxacin are approved for the treatment of columnaris and MAS in aquatic animals (Baoprasertkul, Somsiri & Boonyawiwat, 2012). Mutations in quinolone resistance determining regions (QRDRs) in the gyrA and parC regions, and plasmid-encoded quinolone resistance (PMQR) genes can confer quinolone resistance. However, the main mechanism conferred on resistance in aquatic bacteria has not been clearly investigated. Few studies of the phenotype, genotype, and its determinants of AMR related to foodborne bacteria in hybrid red tilapia have been reported in Thailand. Therefore, the aims of this study were to characterize the phenotype and genotype of AMR bacteria, virulence genes, and ESBL production of A. hydrophila, Salmonella spp., and V. cholerae isolated from the surface of fish, intestine, muscle, liver and kidney of hybrid red tilapia and their cultivation water, and to characterize gyrA and parC in QRDRs of ciprofloxacin-resistant isolates.

Materials & Methods

Bacterial strains

A total of 278 isolates of A. hydrophila (n = 15), Salmonella (n = 188) and V. cholerae (n = 75) were obtained from an existing bank of bacterial isolates from a previous study on bacterial pathogens from cage cultured tilapia in Thailand (Thaotumpitak et al., 2022), which were stored at the Department of Veterinary Public Health, Faculty of Veterinary Science, Chulalongkorn University (Table 1). In summary, these isolates were collected from healthy fresh tilapia (carcass rinse, intestine, muscle, liver, and kidney) and cultivation water in Kanchanaburi Province from October 2019 to November 2020, with the study location and bacterial isolation procedures described previously (Thaotumpitak et al., 2022). All isolates had been stored in 20% glycerol at −80 °C. This study was reviewed and approved by IBC No. 2031027 of the Faculty of Veterinary Science, Chulalongkorn University.

Table 1 Number of bacterial isolates tested in this study (n = 278).

Sample type	No. of bacterial isolate ( n )	
	A. hydrophila	Salmonella	V. cholerae	
Hybrid red tilapia				
Carcass rinse	10	24	10	
Intestine	0	57	23	
Meat	0	0	2	
Liver and kidney	0	1	6	
Cultivation water	5	106	34	
Total	15	188	75	

DNA preparation and polymerase chain reaction (PCR)

The DNA template was prepared using the whole cell boiling method (Lévesque et al., 1995). Briefly, the pure bacterial isolate was streaked onto nutrient agar (Difco, Franklin Lakes, NJ, USA) and incubated at 37 °C for 18–24 h. Then a single colony was harvested and transferred to an Eppendorf tube containing 150 µL of rNase-free water. The well-mixed suspension was heated for 10 min and immediately placed on ice. The suspension was centrifuged at 11,000 rpm for 5 min and the supernatant was used as a DNA template.

Bacterial confirmation

Bacterial confirmation was carried out for A. hydrophila, Salmonella, and V. cholerae using genus-specific and species-specific primers listed in Table 2. Confirmation of A. hydrophila was carried out by targeting genus-specific and species-specific primers (aer and ah), respectively. Salmonella isolates were confirmed by detection of the invasion gene (invA), which also served as the Salmonella virulence gene. The V. cholerae isolates were confirmed using a pair of species-specific primers (ompW) targeting the outer membrane protein. In addition, all V. cholerae isolates were serogrouped by slide-agglutination test with polyvalent V. cholerae O1, monoclonal V. cholerae O139, and monoclonal V. cholerae O141 from commercial antiserum (S&A reagents lab, Bangkok, Thailand).

Table 2 Primers used for bacterial confirmation, AMR genotype, resistance determinants, virulence genes, and QRDRs detection.

Gene	Primer	Oligonucleotide sequences (5′–3′)	Amplicon size (bp)	Reference	
Genus/species confirmation			
A. hydrophila				
aer	aer-F	CTACTTTTGCCGGCGAGCGG	953	Ahmed et al. (2018)	
	aer-R	TGATTCCCGAAGGCACTCCC			
ah	ah-F	GAAAGGTTGATGCCTAATACGTA	625	Ahmed et al. (2018)	
	ah-R	CGTGCTGGCAACAAAGGACAG			
Salmonella				
invA*	invA-F	GTGAAATTATCGCCACGTTCGGGCAA	284	Kumar, Datta & Lalitha (2015)	
	invA-R	TCATCGCACCGTCAAAGGAACC		
V. cholerae				
ompW	ompW-F	CACCAAGAAGGTGACTTTATTGTG	588	Sathiyamurthy, Baskaran & Subbaraj (2013)	
	ompW-R	GAACTTATAACCACCCGCG		
Resistance genes				
bla TEM	blaTEM-F	GCGGAACCCCTATTT	964	Hasman et al. (2005)	
	blaTEM-R	TCTAAAGTATATATGAGTAAACTTGGTCTGAC		
bla SHV	blaSHV-F	TTCGCCTGTGTATTATCTCCCTG	854	Hasman et al. (2005)	
	blaSHV-R	TTAGCGTTGCCAGTGYTG		
bla CTX−M	blaCTX−M-F	CGATGTGCAGTACCAGTAA	585	Batchelor et al. (2005)	
	blaCTX−M-R	AGTGACCAGAATCAGCGG		
bla NDM	blaNDM-F	GGTTTGGCGATCTGGTTTTC	621	Pungpian et al. (2021)	
	blaNDM-R	CGGAATGGCTCATCACGATC		
bla PSE	blaPSE-F	GCAAGTAGGGCAGGCAATCA	422	Chuanchuen et al. (2008)	
	blaPSE-R	GAGCTAGATAGATGCTCACAA		
bla OXA	blaOXA-F	ACACAATACATATCAACTTCGC	813	Costa et al. (2006)	
	blaOXA-R	AGTGTGTGTTTAGAATGGTGATC		
sul1	sul1-F	CGGCGTGGGCTACCTGAACG	433	Khan et al. (2019)	
	sul1-R	GCCGATCGCGTGAAGTTCCG		
sul2	sul2-F	CGGCATCGTCAACATAACCT	721	Khan et al. (2019)	
	sul2-R	TGTGCGGATGAAGTCAGCTC		
sul3	sul3-F	CAACGGAAGTGGGCGTTGTGGA	244	Khan et al. (2019)	
	sul3-R	GCTGCACCAATTCGCTGAACG		
qnrA	qnrA-F	AGAGGATTTCTCACGCCAGG	580	Cattoir et al. (2007)	
	qnrA-R	TGCCAGGCACAGATCTTGAC		
qnrB	qnrB-F	GGMATHGAAATTCGCCACTG	264	Cattoir et al. (2007)	
	qnrB-R	TTTGCYGYYCGCCAGTCGAAC		
qnrS	qnrS-F	GCAAGTTCATTGAACAGGGT	428	Cattoir et al. (2007)	
	qnrS-R	TCTAAACCGTCGAGTTCGGCG		
ermB	ermB-F	AGACACCTCGTCTAACCTTCGCTC	640	Raissy et al. (2012)	
	ermB-R	TCCATGTACTACCATGCCACAGG		
dfrA1	dfrA1-F	GGAGTGCCAAAGGTGAACAGC	367	Shahrani, Dehkordi & Momtaz (2014)	
	dfrA1-R	GAGGCGAAGTCTTGGGTAAAAAC		
dfrA12	dfrA12-F	TTCGCAGACTCACTGAGGG	330	Chuanchuen et al. (2008)	
	dfrA12-R	CGGTTGAGACAAGCTCGAAT		
catA	catA-F	CCAGACCGTTCAGCTGGATA	454	Chuanchuen et al. (2008)	
	catA-R	CATCAGCACCTTGTCGCCT		
catB	catB-F	CGGATTCAGCCTGACCACC	461	Chuanchuen et al. (2008)	
	catB-R	ATACGCGGTCACCTTCCTG		
cmlA	cmlA-F	TGGACCGCTATCGGACCG	641	Chuanchuen et al. (2008)	
	cmlA-R	CGCAAGACACTTGGGCTGC		
strA	strA-F	TGGCAGGAGGAACAGGAGG	405	Chuanchuen et al. (2008)	
	strA-R	AGGTCGATCAGACCCGTGC		
strB	strB-F	GGCAGCATCAGCCTTATAATTT	470	Mala et al. (2017)	
	strB-R	GTGGATCCGTCATTCATTGTT		
tetA	tetA-F	GGCGGTCTTCTTCATCATGC	502	Khan et al. (2019)	
	tetA-R	CGGCAGGCAGAGCAAGTAGA		
tetB	tetB-F	CGCCCAGTGCTGTTGTTGTC	615	Khan et al. (2019)	
	tetB-R	CGCGTTGAGAAGCTGAGGTG		
tetD	tetD-F	AAACCATTACGGCATTCTGC	787	Kumai et al. (2005)	
	tetD-R	GACCGGATACACCATCCATC		
addA1	addA1-F	CTCCGCAGTGGATGGCGG	631	Chuanchuen et al. (2008)	
	addA1-R	GATCTGCGCGCGAGGCCA		
addA2	addA2-F	CATTGAGCGCCATCTGGAAT	500	Chuanchuen et al. (2008)	
	addA2-R	ACATTTCHCTCATCGCCGGC		
aac(3)IV	aac(3)IV-F	GTGTGCTGCTGGTCCACAGC	627	Stoll et al. (2012)	
	aac(3)IV-R	AGTTGACCCAGGGCTGTCGC		
aac(6′)-Ib	aac(6′)-Ib-F	TTGCGATGCTCTATGAGTGGCTA	482	Park et al. (2006)	
	aac(6′)-lb-R	CTCGAATGCCTGGCGTGTTT		
qepA	qepA- F	GCAGGTCCAGCAGCGGGTAG	199	Yamane et al. (2008)	
	qepA- R	CTTCCTGCCCGAGTATCGTG		
floR	floR-F	ATGGTGATGCTCGGCGTGGGCCA	800	Ying et al. (2019)	
	floR-R	GCGCCGTTGGCGGTAACAGACACCGTGA		
mcr-1	mcr-1-F	AGTCCGTTTGTTCTTGTGGC	320	Rebelo et al. (2018)	
	mcr-1-R	AGATCCTTGGTCTCGGCTTG		
mcr-2	mcr-2-F	CAAGTGTGTTGGTCGCAGTT	715	Rebelo et al. (2018)	
	mcr-2-R	TCTAGCCCGACAAGCATACC		
mcr-3	mcr-3-F	AAATAAAAATTGTTCCGCTTATG	929	Rebelo et al. (2018)	
	mcr-3-R	AATGGAGATCCCCGTTTTT		
mcr-4	mcr-4-F	TCACTTTCATCACTGCGTTG	1116	Rebelo et al. (2018)	
	mcr-4-R	TTGGTCCATGACTACCAATG		
mcr-5	mcr-5-F	ATGCGGTTGTCTGCATTTATC	1644	Rebelo et al. (2018)	
	mcr-5-R	TCATTGTGGTTGTCCTTTTCTG		
Integrons and integrative and conjugative elements			
int1	int1-F	CCTGCACGGTTCGAATG	497	Kitiyodom et al. (2010)	
	int1-R	TCGTTTGTTCGCCCAGC		
int2	int2-F	GGCAGACAGTTGCAAGACAA	247	Kitiyodom et al. (2010)	
	int2-R	AAGCGATTTTCTGCGTGTTT		
int3	int3-F	CCGGTTCAGTCTTTCCTCAA	155	Kitiyodom et al. (2010)	
	int3-R	GAGGCGTGTACTTGCCTCAT		
int SXT	intSXT-F	GCTGGATAGGTTAAGGGCGG	592	Kitiyodom et al. (2010)	
	intSXT-R	CTCTATGGGCACTGTCCACATTG		
Virulence genes			
A. hydrophila				
aero	aero-F	CACAGCCAATATGTCGGTGAAG	326	Ahmed et al. (2018)	
	aero-R	GTCACCTTCTCGCTCAGGC		
hly	hly-F	CTATGAAAAAACTAAAAATAACTG	1500	Ahmed et al. (2018)	
	hly-R	CAGTATAAGTGGGGAAATGGAAAG		
V. cholerae				
hlyA	hlyA-F	GGCAAACAGCGAAACAAATACC	481	Imani et al. (2013)	
	hlyA-R	CTCAGCGGGCTAATACGGTTTA		
ctx	ctx-F	CAGTCAGGTGGTCTTATGCCAAGAGG	167	Wong, You & Chen (2012)	
	ctx-R	CCCACTAAGTGGGCACTTCTCAAACT		
tcpA	tcpA-F	CACGATAAGAAAACCGGTCAAGAG	453	Singh, Isac & Colwell (2002)	
	tcpA-R	CGAAAGCACCTTCTTTCACGTTG		
QRDR	
gyrA	gyrA-F	GCTGAAGAGCTCCTATCTGG	436	Chuanchuen & Padungtod (2009)	
	gyrA-R	GGTCGGCATGACGTCCGG		
parC	parC-F	GTACGTGATCATGGATCGTG	390	Chuanchuen & Padungtod (2009)	
	parC-R	TTCCTGCATGGTGCCGTCG			
Notes.

* Virulence gene of Salmonella.

Antimicrobial susceptibility test

The agar dilution method was used to determine antimicrobial susceptibility to twelve antimicrobial agents, including ampicillin, chloramphenicol, ciprofloxacin, enrofloxacin, florfenicol, gentamicin, oxolinic acid, oxytetracycline, streptomycin, sulfamethoxazole, tetracycline, and trimethoprim according to the CLSI standard (CLSI, 2015). These antimicrobials were chosen from common antimicrobials used in human and aquatic medicine. The clinical breakpoints and epidemiological cutoff values of A. hydrophila, Salmonella, and V. cholerae were determined (CLSI, 2014; CLSI, 2015; CLSI, 2016; CLSI, 2020). Reference strains, including Staphylococcus aureus ATCC 29213, Escherichia coli ATCC 25922, and Pseudomonas aeruginosa ATCC 27853 were used for quality control.

Detection of ESBL production

The disc diffusion method was performed according to the CLSI guideline (CLSI, 2015). In the screening test, three disks of ceftazidime (30 µg), cefotaxime (30 µg), and cefpodoxime (10 µg) (Oxoid, Basingstoke, Hampshire, UK) were used to test antimicrobial susceptibility. Isolates that exhibited resistance to at least one of these cephalosporins were further confirmed using a combination disk diffusion method. Ceftazidime (30 µg), cefotaxime (30 µg), and these two disks combined with clavulanic acid were used. The difference in inhibition zone between single cephalosporin and cephalosporins containing clavulanic acid greater than five mm indicates positive ESBL-producing isolates.

AMR gene and virulence-encoded gene detection

All primers for resistance genes, their determinants, and virulence genes are listed in Table 2. Resistance genes were chosen to correspond with AMR phenotypes: blaTEM, blaSHV, blaCTX−M and blaPSE corresponding to β-lactam resistance and ESBL production; blaNDM and blaOXA corresponding to carbapenem resistance; catA, catB, floR and cmlA corresponding to phenicol resistance; ermB corresponding to erythromycin resistance; qnrA, qnrB, qnrS, aac(6′)-Ib, and qepA corresponding to quinolone resistance; aadA1, aadA2 and aac(3)IV corresponding to gentamicin resistance; tetA, tetB and tetD corresponding to tetracycline resistance; strA and strB corresponding to streptomycin resistance; sul1, sul2, and sul3 corresponding to sulfonamide resistance; dfrA1 and dfrA12 corresponding to trimethoprim resistance; mcr-1 to mcr-5 corresponding to colistin resistance. To confirm the existence of colistin-resistant genes (n = 3), sequence-positive mcr isolates from a previous study were used as reference strains (Pungpian et al., 2021). Integrons (int1, int2, and int3) and integrative and conjugative elements (SXT element; intSXT) were also detected. For the presence of virulence genes, isolates of A. hydrophila (aerolysin gene: aero; hemolysin gene: hly), Salmonella (invA: invasive gene) and V. cholerae (hemolysin gene: hlyA, cholera toxin: ctx, and co-regulated toxin: tcpA) were examined to determine their virulence factors.

The final volume (50 µL) of the PCR mixture was prepared according to the manufacturer’s instructions. A 5 µL of template DNA, 25 µL of TopTaq Master Mix (Qiagen, Hilden, Germany), 2 µL of each forward and reverse primer, 5 µL of coralLoad, and 11 µL of sterile rNase free water were used. PCR amplification was carried out on a Tpersonal combi model (Biometra, Göttingen, Germany). The PCR product was separated on 1.5% (w/v) agarose gel and stained with Redsafe™ nucleic acid staining solution (Intron Biotechnology, Seongnam, Republic of Korea). The results were photographed using the Omega Fluor™ gel documentation system (Aplegen, Pleasanton, CA, USA).

Determination of QRDR nucleotide sequences

Salmonella isolates (n = 11) were DNA sequenced for detection of the QRDRs. Two target sites at gyrA and parC were amplified by PCR using the primers listed in Table 2. Eight Salmonella isolates were randomly selected from the intestinal tract (n = 4), cultivation water (n = 2), carcass rinses (n = 2) based on the resistance gradient of ciprofloxacin, and susceptible isolates to ciprofloxacin (n = 3) were used as a negative control. The gyrA and parC were submitted for purification and nucleotide sequencing (Bionics Co., LTD., Gyeonggi-Do, Republic of Korea). The identity of amino acid sequences were analyzed using Molecular Evolutionary Genetic Analysis (MEGA) software version 11 (Tamura, Stecher & Kumar, 2021). Reference sequences (accession number NC_003197:2) from the GenBank database are available at the National Centre for Biotechnology Information (NCBI) (https://www.ncbi.nlm.nih.gov).

Statistical analysis

Descriptive statistics were used to characterize the prevalence of AMR, MDR, virulence genes, integrons, SXT element, ESBL production, and QRDR mutations from A. hydrophila, Salmonella, and V. cholerae isolates. Logistic regression was used to determine the association between resistance and its determinants, virulence factors, and ESBL production (OR >1: positive association; OR <1: negative association). Two-sided hypothesis testing, with p < 0.05 were considered statistically significant. All statistical analyzes were performed with Stata version 14.0 (StataCorp, College Station, TX, USA).

Results

Phenotype and genotype of AMR and virulence genes of A. hydrophila

A. hydrophila was only detected in carcass rinses and cultivation water (Table 1). All isolates (n = 15) were resistant to ampicillin (Table 3), with four isolates (26.7%) resistant to oxytetracycline, tetracycline, and trimethoprim; three isolates (20.0%) were resistant to oxolinic acid. Resistance to at least one antimicrobial and MDR in A. hydrophila were 100% and 26.7%, respectively. Among the six AMR patterns, AMP alone (53.3%) was the most common pattern, with AMP-OTC-TET-TRI and AMP-OXO found in two isolates (13.3%) (Table 4). The most prevalent resistance gene found in A. hydrophila was mcr-3 (20.0%), followed by 13.3% of isolates having floR, qnrS, sul1, sul2, and dfrA1 (Table 3). The mcr-3 was only detected in fish carcass rinses, and only one isolate (6.7%) was positive for int1. The presence of aero and hly was observed in all isolates of A. hydrophila (100.0%) (Table 3).

Table 3 Phenotypic and genotypic resistance and virulence genes of A. hydrophila (n = 15) isolated from hybrid red tilapia carcass rinses and cultivation water.

Variable	Resistance (%)	
	Carcass rinse (n = 10)	Cultivation water (n = 5)	Grand total (n = 15)	
Antimicrobials				
Ampicillin	10 (100.0)	5 (100.0)	15 (100.0)	
Cefotaxime	0 (0)	0 (0)	0 (0)	
Cefpodoxime	0 (0)	0 (0)	0 (0)	
Ceftazidime	0 (0)	0 (0)	0 (0)	
Chloramphenicol	0 (0)	0 (0)	0 (0)	
Ciprofloxacin	0 (0)	0 (0)	0 (0)	
Enrofloxacin	0 (0)	0 (0)	0 (0)	
Florfenicol	0 (0)	0 (0)	0 (0)	
Gentamicin	0 (0)	0 (0)	0 (0)	
Oxolinic acid	3 (30.0)	0 (0)	3 (20.0)	
Oxytetracycline	3 (30.0)	1 (20.0)	4 (26.7)	
Streptomycin	0 (0)	0 (0)	0 (0)	
Sulfamethoxazole	0 (0)	0 (0)	0 (0)	
Tetracycline	3 (30.0)	1 (20.0)	4 (26.7)	
Trimethoprim	3 (30.0)	1 (20.0)	4 (26.7)	
MDR	3 (30.0)	1 (20.0)	4 (26.7)	
AMR genes				
floR	2 (20.0)	0 (0)	2 (13.3)	
qnrS	2 (20.0)	0 (0)	2 (13.3)	
sul1	2 (20.0)	0 (0)	2 (13.3)	
sul2	2 (20.0)	0 (0)	2 (13.3)	
dfrA1	2 (20.0)	0 (0)	2 (13.3)	
mcr-3	3 (30.0)	0 (0)	3 (20.0)	
Integrons				
int1	1 (10.0)	0 (0)	1 (6.7)	
Virulence genes				
aero	10 (100.0)	5 (100.0)	15 (100.0)	
hly	10 (100.0)	5 (100.0)	15 (100.0)	
Notes.

The table presented for positive isolates, which were only detected in carcass rinses and cultivation water. Non-detected genes in A. hydrophila were: blaTEM, blaSHV, blaCTX−M, blaNDM, blaPSE, blaOXA, sul3, qnrA, qnrB, ermB, dfrA12, catA, catB, cmlA, strA, strB, tetA, tetB, tetD, addA1, addA2, aac(3)IV, aac(6′)-Ib, qepA, mcr-1, mcr-2, mcr-4, mcr-5, int2, int3, and intSXT.

Table 4 Resistance patterns of A. hydrophila isolated from hybrid red tilapia carcass rinses and cultivation water (n = 15).

AMR pattern	Resistance (%)	
	Carcass rinses (n = 10)	Cultivation water (n = 5)	Grand total (n = 15)	
AMP	4 (40.0)	4 (80.0)	8 (53.3)	
AMP-OTC-TET-TRI	1 (10.0)	1 (20.0)	2 (13.3)	
AMP-OTC-TRI	1 (10.0)	0 (0)	1 (6.7)	
AMP-OTC-OXO-TET	1 (10.0)	0 (0)	1 (6.7)	
AMP-OXO	2 (20.0)	0 (0)	2 (13.3)	
AMP-TRI	1 (10.0)	0 (0)	1 (6.7)	
Notes.

AMP ampicillin

OTC oxytetracycline

OXO oxolinic acid

TET tetracycline

TRI trimethoprim

Phenotype and genotype of AMR, and virulence genes of Salmonella spp.

All Salmonella isolates (n = 188) were resistant to at least one antimicrobial (Table 5). There was a high prevalence of resistance to ampicillin (79.3%), oxolinic acid (75.5%), oxytetracycline (71.8%) and tetracycline (70.7%). All Salmonella isolates were sensitive to gentamicin (100%) and only two isolates (1.1%) were resistant to trimethoprim. More than 70% of Salmonella isolates exhibited MDR: of the 34 unique resistance patterns, the two most common were AMP-CHP-FFC-OTC-OXO-TET (20.7%) and AMP-CHP-ENR-FFC-OTC-OXO-TET (18.6%) (Table 6). Most Salmonella harbored qnrS (65.4%), followed by tetA (64.9%), blaTEM (63.8%), and floR (55.9%) (Table 5). None of the colistin resistance genes was observed in the Salmonella isolates. In general, cultivation water exhibited higher resistance levels than any other types of fish organ or tissue sample. Based on the type of fish sample, Salmonella isolated from the intestine had a higher prevalence of the AMR phenotype and genotype than the fish carcass rinses. The invA gene was detected in all Salmonella isolates.

Table 5 Phenotypic and genotypic resistance and virulence genes of Salmonella isolates (n = 188) from hybrid red tilapia and cultivation water.

Variable	Resistance (%)	
	Carcass rinse (n = 24)	Intestine (n = 57)	Liver and kidney (n = 1)	Cultivation water (n = 106)	Grand total (n = 188)	
Antimicrobials						
Ampicillin	13 (54.2)	38 (66.7)	1 (100.0)	97 (91.5)	149 (79.3)	
Cefotaxime	0 (0)	0 (0)	0 (0)	0 (0)	0 (0)	
Cefpodoxime	0 (0)	0 (0)	0 (0)	0 (0)	0 (0)	
Ceftazidime	0 (0)	0 (0)	0 (0)	0 (0)	0 (0)	
Chloramphenicol	11 (45.8)	28 (49.1)	1 (100.0)	78 (73.6)	118 (62.8)	
Ciprofloxacin	3 (12.5)	14 (24.6)	0 (0)	23 (21.7)	40 (21.3)	
Enrofloxacin	6 (25.0)	20 (35.1)	0 (0)	43 (40.6)	69 (36.7)	
Florfenicol	12 (50.0)	22 (38.6)	1 (100.0)	69 (65.1)	104 (55.3)	
Gentamicin	0 (0)	0 (0)	0 (0)	0 (0)	0 (0)	
Oxolinic acid	16 (66.7)	40 (70.2)	0 (0)	86 (81.1)	142 (75.5)	
Oxytetracycline	13 (54.2)	34 (59.6)	0 (0)	88 (83.0)	135 (71.8)	
Streptomycin	0 (0)	2 (3.5)	0 (0)	2 (1.9)	4 (2.1)	
Sulfamethoxazole	0 (0)	1 (1.8)	0 (0)	3 (2.8)	4 (2.1)	
Tetracycline	13 (54.2)	32 (56.1)	0 (0)	88 (83.0)	133 (70.7)	
Trimethoprim	0 (0)	0 (0)	0 (0)	2 (1.9)	2 (1.1)	
MDR	13 (54.2)	34 (59.6)	0 (0)	89 (84.0)	136 (72.3)	
AMR genes						
bla TEM	10 (41.7)	34 (59.6)	1 (100.0)	75 (70.8)	120 (63.8)	
floR	10 (41.7)	24 (42.1)	0 (0)	71 (67.0)	105 (55.9)	
qnrS	13 (54.2)	31 (54.4)	0 (0)	79 (74.5)	123 (65.4)	
tetA	13 (54.2)	31 (54.4)	0 (0)	78 (73.6)	122 (64.9)	
tetB	0 (0)	1 (1.8)	0 (0)	2 (1.9)	3 (1.6)	
strA	0 (0)	1 (1.8)	0 (0)	2 (1.9)	3 (1.6)	
sul1	0 (0)	0 (0)	0 (0)	2 (1.9)	2 (1.1)	
sul2	0 (0)	1 (1.8)	0 (0)	2 (1.9)	3 (1.6)	
Virulence genes						
invA	24 (100.0)	57 (100.0)	1 (100.0)	106 (100.0)	188 (100.0)	
Notes.

This table presented only positive isolates from various tilapia samples except fish muscle, which were negative for Salmonella. Non-detected genes in Salmonella were: blaSHV, blaCTX−M, blaNDM,blaPSE, blaOXA, sul3, qnrA, qnrB, ermB, dfrA1, dfrA12, catA, catB, cmlA, strB, tetD, addA1, addA2, aac(3)IV, aac(6′)-Ib, qepA, floR, mcr-1, mcr-2, mcr-3, mcr-4, mcr-5, int1, int2, int3, and intSXT.

Table 6 Resistance patterns of Salmonella isolates (n = 188) from hybrid red tilapia carcass rinses, intestine, liver and kidney, and cultivation water.

AMR pattern	No of isolates (%)	
	Fish carcassrinse (n = 24)	Intestine (n = 57)	Liver andkidney (n = 1)	Cultivation water (n = 106)	Total (n = 188)	
Susceptible	8 (33.3)	11 (19.3)	0 (0)	4 (3.8)	23 (12.2)	
AMP-CHP-CIP-ENR-FFC-OTC-OXO-TET	1 (4.2)	2 (3.5)	0 (0)	1 (0.9)	4 (2.1)	
AMP-CHP-CIP-ENR-OTC-OXO-TET	0 (0)	4 (7.0)	0 (0)	6 (5.7)	10 (5.3)	
AMP-CHP-CIP-FFC-OTC-OXO-TET	0 (0)	0 (0)	0 (0)	3 (2.8)	3 (1.6)	
AMP-CHP-CIP-OTC-OXO-TET	0 (0)	0 (0)	0 (0)	2 (1.9)	2 (1.1)	
AMP-CHP-ENR-FFC-OTC-OXO-STR-TET	0 (0)	1 (1.8)	0 (0)	0 (0)	1 (0.5)	
AMP-CHP-ENR-FFC-OTC-OXO-SMZ-TET	0 (0)	0 (0)	0 (0)	1 (0.9)	1 (0.5)	
AMP-CHP-ENR-FFC-OTC-OXO-TET	5 (20.8)	5 (8.8)	0 (0)	25 (23.6)	35 (18.6)	
AMP-CHP-FFC	0 (0)	3 (5.3)	1 (100.0)	8 (7.5)	12 (6.4)	
AMP-CHP-FFC-OTC	0 (0)	0 (0)	0 (0)	1 (0.9)	1 (0.5)	
AMP-CHP-FFC-OTC-OXO	0 (0)	1 (1.8)	0 (0)	0 (0)	1 (0.5)	
AMP-CHP-FFC-OTC-OXO-TET	5 (20.8)	7 (12.3)	0 (0)	27 (25.5)	39 (20.7)	
AMP-CHP-FFC-OXO-STR	0 (0)	1 (1.8)	0 (0)	0 (0)	1 (0.5)	
AMP-CHP-FFC-TET	0 (0)	0 (0)	0 (0)	3 (2.8)	3 (1.6)	
AMP-CHP-OTC-OXO-TET	0 (0)	1 (1.8)	0 (0)	1 (0.9)	2 (1.1)	
AMP-CHP-OXO	0 (0)	1 (1.8)	0 (0)	0 (0)	1 (0.5)	
AMP-CHP-OXO-TET	0 (0)	1 (1.8)	0 (0)	0 (0)	1 (0.5)	
AMP-CIP-ENR-OTC-OXO-SMZ-TET	0 (0)	1 (1.8)	0 (0)	0 (0)	1 (0.5)	
AMP-CIP-ENR-OTC-OXO-TET	0 (0)	6 (10.5)	0 (0)	8 (7.5)	14 (7.4)	
AMP-CIP-ENR-OTC-OXO-TET-TRI	0 (0)	0 (0)	0 (0)	1 (0.9)	1 (0.5)	
AMP-CIP-FFC-OTC-OXO-TET	1 (4.2)	0 (0)	0 (0)	0 (0)	1 (0.5)	
AMP-CIP-OTC-OXO-TET	1 (4.2)	1 (1.8)	0 (0)	1 (0.9)	3 (1.6)	
AMP-CIP-TET	0 (0)	0 (0)	0 (0)	1 (0.9)	1 (0.5)	
AMP-ENR-OTC-OXO-TET	0 (0)	1 (1.8)	0 (0)	0 (0)	1 (0.5)	
AMP-FFC-OTC-OXO-TET	0 (0)	1 (1.8)	0 (0)	0 (0)	1 (0.5)	
AMP-OTC-OXO-SMZ-STR-TET	0 (0)	0 (0)	0 (0)	1 (0.9)	1 (0.5)	
AMP-OTC-OXO-TET-TRI	0 (0)	0 (0)	0 (0)	1 (0.9)	1 (0.5)	
AMP-OTC-SMZ-STR-TET	0 (0)	0 (0)	0 (0)	1 (0.9)	1 (0.5)	
AMP-OTC-OXO-TET	0 (0)	0 (0)	0 (0)	5 (4.7)	5 (2.7)	
AMP-OTC-TET	0 (0)	1 (1.8)	0 (0)	0 (0)	1 (0.5)	
CHP-FFC	0 (0)	1 (1.8)	0 (0)	0 (0)	1 (0.5)	
ENR-OTC-OXO	0 (0)	0 (0)	0 (0)	1 (0.9)	1 (0.5)	
OTC	0 (0)	1 (1.8)	0 (0)	2 (1.9)	3 (1.6)	
OTC-OXO	0 (0)	1 (1.8)	0 (0)	0 (0)	1 (0.5)	
OXO	3 (12.5)	5 (8.8)	0 (0)	2 (1.9)	10 (5.3)	
Notes.

This table presented only positive isolates from various tilapia samples except fish muscle, which were negative for Salmonella.

AMP ampicillin

CHP chloramphenicol

CIP ciprofloxacin

ENR enrofloxacin

FFC florfenicol

OTC oxytetracycline

OXO oxolinic acid

STR streptomycin

SMZ sulfamethoxazole

TET tetracycline

TRI trimethoprim

Phenotype and genotype of AMR and virulence genes of V. cholerae

All V. cholerae isolates (n = 75) were non-agglutinating Vibrios (NAGs) with O1 and O139, and non-agglutination with O141. All V. cholerae isolates (100%) were susceptible to the 12 antimicrobials tested. The predominant AMR gene in V. cholerae was sul1 (12.0%) (Table 7). The genes catB, qnrS, tetA, tetB, strA, and dfrA1 were all present at a 4.0% prevalence. Colistin resistance genes, integrons, and SXT element were not found in any V. cholerae isolates in this study, in contrast, all isolates had detectable hlyA.

Table 7 Genotypic resistance and virulence genes of V. cholerae isolates (n = 75) of hybrid red tilapia and cultivation water.

Genotype	Prevalence (%)	
	Fish carcassrinse (n = 10)	Intestine (n = 23)	Meat (n = 2)	Liver andkidney (n = 6)	Cultivation water (n = 34)	Total (n = 75)	
AMR genes							
catB	1 (10.0)	0 (0)	0 (0)	0 (0)	2 (5.9)	3 (4.0)	
qnrS	1 (10.0)	0 (0)	0 (0)	0 (0)	2 (5.9)	3 (4.0)	
tetA	1 (10.0)	0 (0)	0 (0)	0 (0)	2 (5.9)	3 (4.0)	
tetB	0 (0)	1 (4.3)	0 (0)	0 (0)	2 (5.9)	3 (4.0)	
strA	0 (0)	1 (4.3)	0 (0)	0 (0)	2 (5.9)	3 (4.0)	
sul1	1 (10.0)	0 (0)	0 (0)	0 (0)	8 (23.5)	9 (12.0)	
dfrA1	1 (10.0)	0 (0)	0 (0)	0 (0)	2 (5.9)	3 (4.0)	
Virulence genes							
hlyA	10 (100.0)	23 (100.0)	2 (100.0)	6 (100.0)	34 (100.0)	75 (100.0)	
ctx	0 (0)	0 (0)	0 (0)	0 (0)	0 (0)	0 (0)	
tcpA	0 (0)	0 (0)	0 (0)	0 (0)	0 (0)	0 (0)	
Notes.

This table showed only positive isolates. Non-detected genes in V. cholerae were: blaTEM, blaSHV, blaCTX−M, blaNDM,blaPSE, blaOXA, sul2, sul3, qnrA, qnrB, ermB, dfrA12, catA, cmlA, strB, tetD, addA1, addA2, aac(3)IV, aac(6′)-Ib, qepA, floR, mcr-1, mcr-2, mcr-3, mcr-4, mcr-5, int1, int2, int3, and intSXT.

ESBL production in A. hydrophila, Salmonella, and V. cholerae

ESBL production was not detected in any of the bacterial strains from tilapia aquaculture or the fish tissues or organs (n = 278).

Association between phenotypic and genotypic AMR

Logistic regression analyses indicated there were statistically significant associations between phenotypic and genotypic resistance of tetracycline and the presence of tetA (O.R. = 259.0, CI [52.3–1283.6], p < 0.0001), sulfamethoxazole resistance and the presence of sul2 (O.R. = 90.3, CI [5.6–1455.1], p = 0.001), and streptomycin resistance and the presence of strA (O.R. = 273.0, CI [1.8–42,372.4], p = 0.029).

Mutation of QRDRs

Eight isolates from the 40 Salmonella isolates that were resistant to ciprofloxacin were randomly selected for sequence analysis and QRDR determination. The serovars of these eight isolates were Virchow (n = 1), Saintpaul (n = 5), Neukoelln (n = 1) and Chartes (n = 1) (Table 8). Six of the eight ciprofloxacin-resistant isolates had a point mutation in gyrA from C to A at position 248 (Ser83Tyr); the other two resistant isolates did not have gyrA mutations. The gyrA sequences from the eight Salmonella isolates have been deposited at GenBank under accession numbers OP831158–OP831165. All isolates carried qnrS without any mutations in the parC region.

Table 8 Mutations of gyrA in QRDRs in ciprofloxacin-resistant Salmonella isolates (n = 8).

Sample type	Serovar ( n )	gyrA mutation	PMQR	MIC (µg/mL)	
				CIP	ENR	OXO	
Carcass rinse	Virchow (1)	–	qnrS	4	4	8	
	Saintpaul (1)	C248A	qnrS	4	2	16	
Intestine	Saintpaul (4)	C248A	qnrS	4–8	2–8	4–32	
Cultivation water	Neukoelln (1)	–	qnrS	4	2	4	
	Chartes (1)	C248A	qnrS	>32	>64	>128	
Notes.

CIP ciprofloxacin

ENR enrofloxacin

OXO oxolinic acid

Discussion

AMR and virulence genes of A. hydrophila

The low prevalence of A. hydrophila observed in this study was consistent with previous work which found a low prevalence of 2.0% and 2.7% in the internal organs and fish muscle, respectively (Ahmed et al., 2018). This low prevalence is likely due to the samples being collected from clinically healthy hybrid red tilapia. Furthermore, fish liver is typically the main target organ for colonization by A. hydrophila; however, none of the liver samples in this study were positive for this bacterium (Algammal et al., 2020). In this study, all isolates (100%) were resistant to ampicillin, which was consistent with previous studies in tilapia and other aquatic animals due to their intrinsic resistance (El-ghareeb, Zahran & Abd-Elghany, 2019). More than a fourth of the A. hydrophila isolates (26.7%) were resistant to oxytetracycline, tetracycline, and trimethoprim, which was also consistent with previous studies (El-ghareeb, Zahran & Abd-Elghany, 2019; Yu et al., 2021). The wide range resistance of A. hydrophila observed in this study could result from these bacteria persisting in high concentrations of antimicrobials leading to up-regulation of the efflux pump and the production of antioxidative agents (Yu et al., 2021).

Oxytetracycline and oxolinic acid are broad-spectrum antimicrobials belonging to tetracyclines and fluoroquinolones, which have been approved for use in aquatic animals (OIE, 2018). These antimicrobials were the priority antibiotics in use when outbreaks occurred on the hybrid red tilapia farms enrolled in this study, possibly leading to widespread selection of resistant A. hydrophila. A previous study found that A. hydrophila can harbor various tet genes and transfer resistance to tetracycline to E. coli (Harnisz, Korzeniewska & Gołaś, 2015). Therefore, A. hydrophila might serve as an important reservoir of resistance to tetracycline in the aquatic environment. Regarding resistance in the A. hydrophila isolates, all isolates resistant to oxolinic acid were susceptible to both ciprofloxacin and enrofloxacin, although these antimicrobials were grouped in similar quinolone derivatives. Furthermore, a study of Aeromonas isolated from diseased hybrid red tilapia in Thailand also showed a difference in the resistant phenotype (Mursalim et al., 2022). The explanation for this discrimination could be that different generations conferred distinct antimicrobial potency (Sinwat et al., 2018; Pham, Ziora & Blaskovich, 2019). In detail, ciprofloxacin and enrofloxacin are second-generation quinolones, therefore, they have broader bactericidal activities against Gram-negative bacteria compared to oxolinic acid, which is a first-generation quinolone. To quantify the accurate antimicrobial susceptibility testing in bacteria isolated from aquatic animals, oxolinic acid should be taken into account as a mandatory antimicrobial test regardless of whether other quinolones were selected.

In Thailand, colistin has been banned in food-producing animals as a feed additive since 2017; however, therapeutic use of colistin is still being reported in pig farms (Olaitan et al., 2021; Rueanghiran et al., 2022) and may select for colistin-resistance in these livestock facilities. The dissemination of colistin-resistant enteric bacteria from livestock into the environment could accelerate the horizontal gene transfer of mobile genetic elements to autochthonous aquatic bacteria (Elbediwi et al., 2019). The resistance of colistin and mcr genes in Aeromonas was previously reported in healthy freshwater fish and cultivation water in China (Li et al., 2022). This study confirmed that colistin resistant A. hydrophila isolates already existed in hybrid red tilapia aquaculture. To our knowledge, this finding is the first report on the occurrence of mcr-3 in freshwater fish originating in Thailand. The mcr-3 had been previously detected in E. coli and Salmonella in pigs and pork (Pungpian et al., 2021), with the global spread of mcr-1, mcr-3 and mcr-4 is shown in Fig. 1.

In this study, all A. hydrophila isolates contained aero and hly genes, which were more prevalent than a previous study of freshwater fish in Egypt (Ahmed et al., 2018). These virulence genes found in A. hydrophila were also reported in an outbreak of diarrheal patients in Brazil (Silva et al., 2017). These findings suggest that A. hydrophila isolated from hybrid red tilapia and cultivation water can be both resistant and pathogenic which elevates the public health and food safety risk from these bacteria.

AMR and virulence genes of Salmonella

Isolates of Salmonella from tilapia and cultivation water exhibited a high prevalence to ampicillin (79.3%), which may be due to part to intrinsic resistance to penicillins by Salmonella. In addition, amoxicillin has been approved for off-label use to treat streptococcosis in tilapia aquaculture by the Thai Food and Drug Administration (Baoprasertkul, Somsiri & Boonyawiwat, 2012). The extensive use of amoxicillin in Thai aquaculture can select for β-lactam-resistant bacteria that then colonize tilapia and the immediate aquaculture environment. This study also found high Salmonella resistance to oxolinic acid and oxytetracycline. The improper use of oxolinic acid and oxytetracycline can select for the development of AMR and MDR Salmonella. In particular, high resistance to chloramphenicol (62.8%) and florfenicol (55.3%) was observed, although antimicrobials in the amphenicol groups have not been approved for use in aquaculture in Thailand. Among this collection of isolates, resistance to gentamicin and trimethoprim were not observed which was similar to a previous study in Kenya (Wanja et al., 2020). Moreover, the high prevalence (>75%) of MDR Salmonella isolates is consistent with previous studies in freshwater fish (Saharan, Verma & Singh, 2020; Gawish et al., 2021).

Regarding genotypic AMR, the predominant resistance genes were qnrS (65.4%) which was similar in tilapia and catfish in Egypt (Algammal et al., 2022), and tetA (64.9%) and floR (55.9%) which was also observed in a previous study on farmed fish in Brazil (Ferreira et al., 2021). These phenotypic and genotypic AMR findings indicate a likelihood of transmission of AMR Salmonella by consumption of contaminated fish products.

Figure 1 Global distribution of colistin resistance genes in bacteria isolated from fish.

AMR and virulence genes of V. cholerae

Human infection with V. cholerae is a major public concern because this bacterium is a causative agent of cholera, a severe fatal diarrhea caused by the cholera toxin. Serogroups O1 and O139 have been widely reported in cholera outbreaks with high mortality in humans worldwide. On the contrary, serogroup O141 caused sporadic outbreaks of cholera-like diarrhea (Elimian et al., 2019). All V. cholerae isolates found in this study belonged to non-O1/O139 and non-O141 serogroups, which is consistent with a low prevalence of these serogroups in fish and aquatic environments and in contrast to their high prevalence in human cases (Halpern & Izhaki, 2017; Schwartz et al., 2019). However, the infection of indigenous non-O1/O139 and non-O141 V. cholerae originating from aquaculture products can cause watery-to-severe diarrhea in humans due to other virulence factors (Vezzulli et al., 2020).

Among the V. cholerae isolates in this study, none of them was phenotypically resistant to antimicrobials. This finding is consistent with a previous study that found a low prevalence of AMR in non-cholera environmental strains (Bier et al., 2015). In contrast, a study in China found that over half of V. cholerae (57.6%) isolated from freshwater fish (n = 370) were MDR to streptomycin, ampicillin, and rifampin (Fu et al., 2020). Therefore, it is postulated that resistance to non-used antimicrobials in aquaculture may result from the distribution of AMR bacteria from the anthropogenic sources that then contaminate the aquaculture environment. For example, wastewater and effluent from human communities could be an important contributor to AMR in aquaculture. The most prevalent AMR genotype of V. cholerae in this study was sul1 (12.0%), which encoded resistance to sulfamethoxazole. This finding could be problematic for human clinical treatment options because sulfamethoxazole is a drug of choice for the treatment of cholera (Leibovici-Weissman et al., 2014). In particular, a recent report showed that V. parahaemolyticus isolated from shrimp in China contained the mcr-1 gene (Lei et al., 2019). These findings highlight the need to monitor these bacteria in aquaculture for the emergence of colistin-resistant genes, especially in non-Enterobacterale.

Among the virulence genes examined in this study, hlyA, ctx, and tcpA, the hlyA genes were the most prevalent (100.0%), which was consistent with a previous study regarding V. cholerae isolated from fish, shellfish, and environmental samples (Shan et al., 2022). The hlyA encoding pore-forming toxin can cause cytotoxicity and cell vacuolation of intestinal cells leading to fluid leakage (Ramamurthy et al., 2020). The common prevalence of the hlyA gene was reported in clinical or epidemic strains of V. cholerae isolates, indicating its likely role in the pathogenesis of many cases of V. cholerae (Imani et al., 2013). No strains carrying ctx or tcpA were detected among V. cholerae isolates in this study. The ctx gene that encodes cholera enterotoxin leads to massive secretion of electrolytes and water into the intestinal lumen leading to severe fluid loss, while the tcpA gene acts as a promotor of pilus formation inducing bacterial colonization in the host’s intestine (Ramamurthy et al., 2020). This indicated that the V. cholerae in this study were not cholera-causing strains; nonetheless, human infection with these V. cholerae strains can pose a risk of diarrhea due to the presence of hlyA.

ESBL production in A. hydrophila, Salmonella, and V. cholerae

In this study, no ESBL-producing pathogenic bacteria were detected in hybrid red tilapia and cultivation water. However, the increasing prevalence of ESBL-producing bacteria has been reported in environmental water, livestock animals and humans in Thailand (Runcharoen et al., 2017; Lay et al., 2020). The finding of ESBL-producing bacteria in aquaculture environments was speculated to be due to bacterial contamination of natural water resources used for aquatic animal cultivation. Several studies have reported that ESBL-producing bacteria are presented in many aquatic animals, such as tilapia, catfish, and shrimp (Hon et al., 2016; Gawish et al., 2021). Importantly, bacteria harboring ESBL genes can co-select other plasmid-mediated AMR, such as sulfamethoxazole/trimethoprim, quinolones, and colistin, which are critically important antimicrobials. Infection of these co-resistant bacteria can hamper disease treatment due to the limitation of available therapeutic options, resulting in longer hospital stays and increased antibiotic costs. The surveillance of ESBL-producing bacteria in hybrid red tilapia helps monitor and quantify the possible novel risk of ESBL transmission through consumption of cultured tilapia.

Association between phenotypic and genotypic AMR

In general, tetA was the predominant genotypic resistance observed in Salmonella and V. cholerae isolates. This gene can confer high phenotypic resistance to tetracycline, which was consistent with the statistical association between phenotypic and genotypic resistance to tetracycline based on the logistic regression analyses in this study. Oxytetracycline is a common antimicrobial used in tilapia farms in Thailand due to its effective treatment of aeromoniasis and francisellosis, which are endemic in Thailand (Baoprasertkul, Somsiri & Boonyawiwat, 2012). Previous study also observed a high prevalence of tetA in freshwater fish and water, and other tet genes, such as tetL, tetO, and tetW were also previously reported in aquaculture (Harnisz, Korzeniewska & Gołaś, 2015). The statistical association between resistance to sulfamethoxazole and the presence of sul2, and streptomycin and the presence of strA indicated that phenotypic resistance is strongly associated with the presence of the corresponding resistance genes. Other pairs of AMR phenotype-genotype did not observe statistical association. This may be due to non-examined genes or other mechanisms conferred to those resistance.

Mutation of QRDRs

Of eight ciprofloxacin resistant Salmonella, only a single-point mutation in gyrA (Ser83Tyr) was observed in six isolates. In Salmonella, mutations in gyrA of QRDRs are a substantial mechanism of resistance to quinolones, while parC mutations are rare (Shaheen et al., 2021). The amino acid change in gyrA was reported in Salmonella isolated from chicken, pork, and clinical isolates of humans (Sinwat et al., 2018). This study was in agreement with the frequent identification of gyrA mutations in environmental Enterobacterale in the aquatic environment, including Salmonella (Johnning et al., 2015). The qnrS, which is the PMQR gene presented in all isolates, can mediate low resistance to quinolone and enhance resistance mediated by gyrA mutations. Higher MICs were mostly observed in mutant isolates. However, three isolates were resistant to ciprofloxacin and oxolinic acid, but susceptible to enrofloxacin (Table 8). The inconsistency of resistance to quinolone has previously reported and is still not conclusive (Sinwat et al., 2018). Two ciprofloxacin-resistant Salmonella isolates were not mutants in the gyrA and parC regions. Other genes that confer resistance to quinolones, including aac(6′)-Ib, the aminoglycoside acetyltransferase variant gene, and qepA, the plasmid-mediated nonspecific efflux pump gene, were absent from these eight ciprofloxacin-resistant Salmonella. These results imply that the mutations in gyrA and qnrS expression were pivotal factors driving resistance in Salmonella. The common use of quinolones in both animals and humans can promote horizontal gene transfer and the acquisition of quinolone-resistance genes between these two sectors through environment links. Mechanisms of quinolone resistance in Salmonella need to be further investigated. The rational use of common antimicrobials and rigorous antimicrobial stewardship for both animal and human therapeutic intervention can delay the spread of AMR. Increasing AMR monitoring and developing AMR control strategies in aquaculture to overcome adverse consequences of AMR should be initiated immediately.

Conclusions

Findings from this study on AMR bacteria in hybrid red tilapia suggest that aquaculture as practiced in Thailand can select for ubiquitous AMR pathogens, mobile genetic elements, and an emerging reservoir of mcr and colistin-resistant bacteria. Resistant and pathogenic bacteria, such as resistance to ampicillin and tetracycline, or MDR Salmonella circulating in aquaculture together highlight the public health concerns and foodborne risks of zoonotic pathogens to humans from cultured freshwater fish. The main sources of AMR bacteria should be evaluated to better understand of their circulation between aquaculture and their production environment. Good personal hygiene, occupational safeguards, and farm biosecurity are highly recommended to reduce AMR bacterial contamination of freshwater fish. The surveillance and monitoring of AMR in aquaculture under One Health, and improved antimicrobial stewardship for farmers, should be promoted to better control AMR in tropical aquaculture.

Supplemental Information

Supplemental Information 1 Raw data

Click here for additional data file.

Supplemental Information 2 Sequencing data

Click here for additional data file.

The authors thank Sutida Chalee and Saran Anuntawirun for field collection and laboratory assistance.

Additional Information and Declarations

Competing Interests

Author Contributions

DNA Deposition

Data Availability

The authors declare there are no competing interests.

Varangkana Thaotumpitak performed the experiments, analyzed the data, prepared figures and/or tables, authored or reviewed drafts of the article, and approved the final draft.

Jarukorn Sripradite performed the experiments, prepared figures and/or tables, authored or reviewed drafts of the article, and approved the final draft.

Edward R. Atwill conceived and designed the experiments, authored or reviewed drafts of the article, and approved the final draft.

Saharuetai Jeamsripong conceived and designed the experiments, performed the experiments, analyzed the data, prepared figures and/or tables, authored or reviewed drafts of the article, and approved the final draft.

The following information was supplied regarding the deposition of DNA sequences:

The sequences of gyrA are available at GenBank: OP831158 to OP831165.

The following information was supplied regarding data availability:

The raw data and sequencing data are available in the Supplementary Files.

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
