# Peer review of "Emergence of colistin resistance and characterization of antimicrobial resistance and virulence factors of Aeromonas hydrophila, Salmonella spp., and Vibrio cholerae isolated from hybrid red tilapia cage culture"

_PeerJ, doi:10.7717/peerj.14896_

## Round 0.1 · original submission · Minor Revisions

Dear Dr. Jeamsripong,

Your manuscript requires a number of Minor Revisions.

·

Basic reporting

The title of this article seems interesting, and the work done is commendable. The overall working pattern is good, verifications through PCRs and according to guidelines per performed, which makes this article authentic but there are many basic mistakes other than methodology. For example, I highly recommend the authors to provide the English certificate after review from an English native.

Experimental design

The experimental design is good, it is according to the guidelines and authors have successfully presented the complexity of the matter, highlighting a vast range of resistances and testing more than enough genes. over all this section seems the strongest of all.

Validity of the findings

If they have 217 isolates, and only 15 Hydrophila, we need to check with the statistician, about the acceptance of the results. I would ask the editor to maybe outsource this job.

Additional comments

I am happy with the overall presentation of this article.

Reviewer 2 ·

Basic reporting

Dear all,
I have no major issue regarding the manuscript.
It is thoroughly written and well discussed. I t easy to be understanded by the readers.

Experimental design

Research questions are well defined and relevant.

Validity of the findings

Conclusions are well stated, linked to original research question & limited to supporting results.

·

Basic reporting

The manuscript by Thaotumpitak et al., titled Emergence of colistin resistance and characterization of antimicrobial resistance and virulence factors of Aeromonas hydrophila, Salmonella spp., and Vibrio cholerae isolated from hybrid red tilapia cage culture has been well designed and written. It addresses the issue of global interest i.e AMR. Of particular interest is the issue of AMR in Pisces which has been seldom reported in the literature, which I think will generate much more interest from the scientific community. However, I have the following observations and comments that would improve the quality of the manuscript.

Abstract: Not well written. The method under this section is not appropriate. It indicated the objective or aim of the study. The authors should briefly describe the methods for the detection and how the analyses were carried out. The result section of this structure abstract is too long and should contain the salient points of your findings. Please reduce the length.
Conclusion: There was a repetition of words in line 55 and the transmission of MDR from aquaculture to humans please reflect as appropriate.
Introduction: Literature review, this is well written but needs to talk of public health significance after consumption of contaminated fish resulting in to non-typhoidal Salmonella-gastroenteritis and or NTS-associated bloodstream infection. See Akinyemi et al, 2020, 2021 on that aspect etc.
Methods: Did the Confirmation of isolates used based on the primer sets of the various pathogens evaluated? Were you able to validate the primer sets used? If yes how? And if No Why? How reliable are they? Hence, the need to discuss the reliability of the gene markers for the detection of the pathogens- For instance invA please on reliability see Sharma I, Das K, 2016, Akinyemi et al., 2021 Kadry et al., 2019
Results
Table 1 showed the list of various primers from published data used. Were you able to confirm the reliability of the primer sets used, though they are from published work? Please provide answers.
Table 3 is not self-explanatory, and the legends have no bearing on the caption in the title of the table and the content of the table as well. It makes no sense to this reviewer, please reconstruct it for better understanding. E.g Carcass and Cultivation water must be captured in the title in relation to the presence or absence of A. hydrophilia. In short, the title is faulty.
Table 4: The title is not explanatory to capture the content, please re-capture.
Tables 5, 6 and 7: The titles do not reflect the contents of these tables at all. Please look at the contents and recapture. Note that you did not only obtain values from Cultivation water, what about other variables that were reflected in the content of these tables? Please address these gaps.
Figure 1: How did the authors come about the values indicated in the map? Because I did not see any form of a systematic review of literature from this present study to have warranted such a map and corresponding values attached to some countries in some continents as well. Please this is to me an overzealous and fictitious assertion, however, I may be proved wrong, but I doubt it.

Discussion
Lines 303 –305; It is not proper to say that studies showed that mcr-3 had been isolated
from E. coli and Salmonella in pigs and pork in Thailand but rather mcr-3 had been detected in E.coli Salmonella etc.
Lines 372 – 374; This indicated that V. cholerae in this study was not cholera-causing
strain. However, it is observed that infection with this V. cholerae can pose a risk of diarrhea due to the presence of hlyA how did you know? Did you perform a pathogenicity test? please clarify.

Lines 390-392: This gene can confer high phenotypic resistance to tetracycline and
oxytetracycline supported by a statistical association between phenotypic and genotypic
resistance to tetracycline based on logistic regression analysis where is the statistical value? Please indicate.
Conclusion: The conclusion is too general. It is not the reflection of the content and findings. This could be improved by taking clues from the conclusion of the abstract and expatiating on that before the general assertion.

Experimental design

The manuscript by Thaotumpitak et al., titled Emergence of colistin resistance and characterization of antimicrobial resistance and virulence factors of Aeromonas hydrophila, Salmonella spp., and Vibrio cholerae isolated from hybrid red tilapia cage culture has been well designed and written.

Validity of the findings

The findings are of interest to Scientific Community. However, the following must be addressed
The Table 1 showed the list of various primers from published data used. Were you able to confirm the reliability of the primer sets used, though they are from published work? Please provide answers.
Table 3 is not self-explanatory, and the legends have no bearing on the caption in the title of the table and the content of the table as well. It makes no sense to this reviewer, please reconstruct it for better understanding. E.g Carcass and Cultivation water must be captured in the title in relation to the presence or absence of A. hydrophila. In short, the title is faulty.
Table 4: The title is not explanatory to capture the content, please re-capture.
Tables 5, 6 and 7: The titles do not reflect the contents of these tables at all. Please look at the contents and recapture. Note that you did not only obtain values from Cultivation water, what about other variables that were reflected in the content of these tables? Please address these gaps.
Figure 1: How did the authors come about the values indicated in the map? Because I did not see any form of a systematic review of literature from this present study to have warranted such a map and corresponding values attached to some countries in some continents as well. Please this is to me an overzealous and fictitious assertion, however, I may be proved wrong, but I doubt it.

Additional comments

Should improve on the conclusion and general grammatical expression consistency

---

## Round 0.2 · accepted · Accept

I am writing to inform you that your manuscript - Emergence of colistin resistance and characterization of antimicrobial resistance and virulence factors of Aeromonas hydrophila, Salmonella spp., and Vibrio cholerae isolated from hybrid red tilapia cage culture - has been Accepted for publication. Congratulations!

·

Basic reporting

I am satisfied with the reply and the changes performed by the authors. I would like to request the editor to proceed with the publishing criteria.

Experimental design

I am satisfied with the reply and the changes performed by the authors. I would like to request the editor to proceed with the publishing criteria.

Validity of the findings

I am satisfied with the reply and the changes performed by the authors. I would like to request the editor to proceed with the publishing criteria.